# Local and Systemic Response to Heterogeneous Sulfate Resupply after Sulfur Deficiency in Rice

**DOI:** 10.3390/ijms23116203

**Published:** 2022-05-31

**Authors:** Ru-Yuan Wang, Li-Han Liu, Fang-Jie Zhao, Xin-Yuan Huang

**Affiliations:** State Key Laboratory of Crop Genetics and Germplasm Enhancement, College of Resources and Environmental Sciences, Nanjing Agricultural University, Nanjing 210095, China; 2017203018@njau.edu.cn (R.-Y.W.); 2019103103@stu.njau.edu.cn (L.-H.L.); fangjie.zhao@njau.edu.cn (F.-J.Z.)

**Keywords:** rice, sulfur, transcriptome, split-root system, systemic signaling

## Abstract

Sulfur (S) is an essential mineral nutrient required for plant growth and development. Plants usually face temporal and spatial variation in sulfur availability, including the heterogeneous sulfate content in soils. As sessile organisms, plants have evolved sophisticated mechanisms to modify their gene expression and physiological processes in order to optimize S acquisition and usage. Such plasticity relies on a complicated network to locally sense S availability and systemically respond to S status, which remains poorly understood. Here, we took advantage of a split-root system and performed transcriptome-wide gene expression analysis on rice plants in S deficiency followed by sulfate resupply. S deficiency altered the expressions of 6749 and 1589 genes in roots and shoots, respectively, accounting for 18.07% and 4.28% of total transcripts detected. Homogeneous sulfate resupply in both split-root halves recovered the expression of 27.06% of S-deficiency-responsive genes in shoots, while 20.76% of S-deficiency-responsive genes were recovered by heterogeneous sulfate resupply with only one split-root half being resupplied with sulfate. The local sulfate resupply response genes with expressions only recovered in the split-root half resupplied with sulfate but not in the other half remained in S deficiency were identified in roots, which were mainly enriched in cellular amino acid metabolic process and root growth and development. Several systemic response genes were also identified in roots, whose expressions remained unchanged in the split-root half resupplied with sulfate but were recovered in the other split-root half without sulfate resupply. The systemic response genes were mainly related to calcium signaling and auxin and ABA signaling. In addition, a large number of S-deficiency-responsive genes exhibited simultaneous local and systemic responses to sulfate resupply, such as the sulfate transporter gene *OsSULTR1;1* and the *O-acetylserine (thiol) lyase* gene, highlighting the existence of a systemic regulation of sulfate uptake and assimilation in S deficiency plants followed by sulfate resupply. Our studies provided a comprehensive transcriptome-wide picture of a local and systemic response to heterogeneous sulfate resupply, which will facilitate an understanding of the systemic regulation of S homeostasis in rice.

## 1. Introduction

S is an essential macronutrient playing an important role in plant growth and development. Plants take up S mainly in the form of inorganic sulfate from soils. In *Arabidopsis thaliana*, the uptake of sulfate is mainly mediated by two root-specific high-affinity sulfate transporters, AtSULTR1;1 and AtSULTR1;2 [1]. Upon entry into the root cells, parts of sulfate are stored in the vacuoles in the roots, and the rest is translocated to the shoots and assimilated into organic sulfur. The root-to-shoot translocation of sulfate requires two sulfate transporters, AtSULTR2;1 and AtSULTR3;5 [2,3]. The reduction and assimilation of sulfate mainly take place in the shoots, which have been well studied [4]. In the first step of assimilation, sulfate is activated by ATP sulfurylase (ATPS) to form adenosine 5′-phosphosulfate (APS). APS can be either reduced to sulfite by APS reductase (APR) or phosphorylated by APS kinase (APK) to form 3′-phosphoadenosine 5′-phosphosulfate (PAPS). In the primary sulfate assimilation branch, sulfite is further reduced to sulfide by sulfite reductase (SiR), while in the secondary assimilation branch, PAPS provides an activated sulfate donor for many sulfation reactions. Sulfide could be further integrated into the skeleton of *O*-acetylserine (OAS) by *O*-acetylserine (thiol) lyase (OAS-TL) to form cysteine (Cys), the first organic-reduced sulfur compound. Cys then serves as a precursor for the biosynthesis of methionine (Met), glutathione (GSH), and numerous S-containing compounds that are essential for plant growth and resistance to stresses. Roots are the main uptake organs of sulfate, whereas most sulfate reduction and assimilation occur in shoots [5,6]. Therefore, the uptake, distribution, and assimilation of sulfate must be tightly controlled in plants, especially under S-limited conditions.

In recent decades, as environmental protection efforts are being intensified, soil S deficiency has become an emerging problem and leads to the deterioration of crop plants [7]. When faced with S deficiency, plants make numerous physiological and morphological changes in order to adapt to a limitation of S [6,8]. Such changes are driven by complex transcriptional regulations in sulfate uptake, assimilation, and metabolism. Numerous S-deficiency-responsive genes have been identified in different species [9,10,11]. Typical S-deficiency-responses in *Arabidopsis* are the activation of sulfate uptake by up-regulating the expressions of high-affinity sulfate transporter genes *AtSULTR1;1* and *AtSULTR1;2* in the roots and the enhancement of sulfate assimilation in the shoots by activating the expressions of *At**ATPS1*, *At**ATPS3*, and *At**ATPS4* [12,13,14]. The root-to-shoot translocation of sulfate mediated by the AtSULTR2 family transporter is enhanced to support the demand for sulfate in the shoots [3,15]. Meanwhile, the efflux of sulfate from vacuoles by AtSULTR4;1 and AtSULTR4;2 are also enhanced under S deficiency conditions [16]. The S containing secondary metabolites, such as glucosinalate in *Arabidopsis*, are degraded to support the demand for S for critical biological processes [17]. Therefore, the ability of plants to respond to S deficiency is therefore essential for adaptation to a soil environment with a limited supply of S. Plants must coordinate the local and systemic signals to optimize the adaptive responses on a whole-plant level under S deficiency. 

A plant’s response to nutrient status depends on the perception of nutrient signals which generally includes local sensing and systemic sensing. The local sensing refers to the perception of external nutrient status in partial tissues such as roots, while the systemic sensing integrates the internal nutrient status in all tissues of plants [18,19]. The split-root system in which the same plants are physically separated into two halves and exposed to heterozygous nutrient status has been used to identify local and systemic sensing and signaling components under various nutrient deficiencies [15,19,20,21]. Under heterozygous S deficiency, the expressions of *AtSULTR1;1* and *AtSULTR1;2* were only induced locally in the S-depleted root half but not in the other root half supplied with S [19]. A similar local but not systemic induction by heterozygous S deficiency was observed for other sulfate transporter genes in *A. thaliana* and *Brassica oleracea* [6,22]. However, the expressions of *MtSULTR1.1* and *MtSULTR1.3* in *Medicago truncatula* were slightly systemically responsive to S deficiency [23,24]. Moreover, the induction of several S-deficiency-responsive genes was attenuated in the sulfate-depleted split-root half compared to the fully sulfate-depleted roots, such as *SDI1* and *LSU1*, *MSA1/SHM7*, and *ChaC-like*, suggesting the existence of systemic regulation of S-deficiency-responsive genes in *A. thaliana* [9,25,26,27,28]. 

The long-distance signal and underlying mechanisms that systemically regulate S deficiency response are not fully understood. The exogenous application of GSH to one side of the roots suppresses the sulfate uptake and ATPS activity in the other untreated split-root half, suggesting that GSH may function as a long-distance signal from the shoots to the roots to negatively regulate S uptake and assimilation [29]. However, by feeding the *apk1 apk2* double mutant with ^35^SO_4_^2−^ and inhibiting the biosynthesis of GSH, Hubberten et al. (2012a) suggested that sulfate in the roots itself but not the GSH from the shoots is the dominant cause of the reduced presence of S in the roots [26]. More studies are required to isolate the systemic signaling molecule(s) that coordinate the S-deficiency response at the whole-plant level.

The complicated and finely tuned response to S deficiency at the local and systemic levels is therefore essential for plants to adapt to S-limited environments. Once supplied with sufficient sulfate, plants grown in S-limited conditions must sense and respond to S resupply in order to recover to a normal growth level. However, the local and systemic response of S-deficient plants to sulfate resupply is largely unknown, especially for crop plants such as rice, a staple crop required for more than half of the world population. In this study, we took advantage of the split-root system and transcriptome analysis to investigate the local and systemic response of S-deficient rice plants to sulfate resupply. We identified a set of genes locally responding to sulfate resupply, including genes involved in amino acid metabolic processes, root growth, and development. The systemic responsive genes were also identified, which were enriched in calcium transport and signaling processes. Our results provide a transcriptome-wide picture of local and systemic responses of S-deficient plants to sulfate resupply in rice, which will facilitate an understanding of the systemic regulation of S homeostasis in rice.

## 2. Results

### 2.1. Experimental Design and RNA-Seq Data Summary

To investigate whether the response of S-deficient rice plants to sulfate resupply is subjected to local and systemic regulation, a hydroponic split-root system was employed (Figure 1A). Plants grown with sufficient sulfate (0.45 mM SO_4_^2−^) to the five-leaf stage were subjected to sulfate deficient treatment without adding sulfate to the nutrient solution for 15 d. The fresh weight of the shoots and roots of the sulfate-deprived plants (termed S0) was dramatically decreased compared to that of the control plants (termed CK) with a sufficient sulfate supply (0.45 mM SO_4_^2−^) (Figure 1B). The total S concentrations in both the shoots and roots of sulfate-deprived plants were also significantly decreased compared to the control plants (Figure 1C,D). These results suggested that the sulfate-deprived plants underwent a strong S deficiency stress. Sulfate was resupplied to the sulfate-deprived plants in the split-root system. One root half was resupplied with sulfate (termed as SPSR for split-root sulfate resupply) for 12 h while the other root half remained under sulfate-deprived conditions (termed as SPS0 for split-root with 0 mM SO_4_^2−^) (Figure 1A). The sulfate-deprived plants in the split-root system were resupplied on both sides (termed as SR for whole roots sulfate resupply) for 12 h as control (Figure 1A). No significant difference in the total S concentrations between the shoots of plants with whole roots sulfate resupply (SR_Sh) and split-root sulfate resupply plants (SPSR_Sh) (Figure 1C). There was also no significant difference in the total S concentrations among the whole roots with sulfate resupply (SR_R), roots of SPSR plants (SPSR_R), and the roots of SPS0 plants (SPS0_R) (Figure 1C). These results indicated that a 12 h sulfate resupply did not significantly change the total S concentrations in sulfate-deprived plants.

To gain insight into the transcriptomic response of rice plants to S deficiency followed by sulfate resupply, RNA sequencing (RNA-seq) was performed on the shoots and roots of rice plants under sufficient sulfate conditions, sulfate deficiency, homogeneous sulfate resupply on both sides of split-roots, and heterogeneous sulfate resupply in only one split-root half. Twenty-seven samples in total were subjected to RNA-seq, including four shoot samples (CK_Sh, S0_Sh, SR_Sh, SP_Sh) and five root samples (CK_R, S0_R, SR_R, SPSR_R, SPS0_R) with three replicates for each of the samples (Figure 1A). The RNA sequencing yielded a total of approximately 14 billion raw reads, with an average of 51.85 million raw reads per sample (Appendix A). After filtering, each library contained approximately 51.76 million clean reads and 77.47 Mb clean bases. The percentages of Q30 bases were in excess of 91.63%, and the GC content of each library was around 55%. On average, 91.78% of clean reads were mapped to the Nipponbare reference genome, with 88.09% being unique mapped and less than 4.55% being multiple mapped, which were excluded from the further analysis (Appendix A). In total, 37,344 transcripts were detected in all samples, with 1678 transcripts being newly annotated transcripts.

The correlation among the three biological replicates in each of the samples was calculated to reflect the data reliability. The correction coefficient among the replicates ranged from 0.8824 to 0.9925 (Appendix A). Furthermore, principal component analysis (PCA) revealed that the three replicates of each treatment were generally grouped together, suggesting that the transcriptome data were reliable with high repeatability (Appendix A).

### 2.2. Transcriptomic Response to S deficiency in Rice

To investigate the transcriptomic response to sulfate deficiency in rice, we compared the FPKM (Fragments Per Kilobase of exon model per Million mapped fragments) values of sulfate-deprived samples (S0) to the control samples (CK). By using absolute fold change ≥ 2 with *p* < 0.05 and a false discovery rate (FDR) < 0.01 as the threshold, 1589 differentially expressed genes (DEGs) were identified in the shoots, including 967 up-regulated genes and 622 down-regulated genes in the shoots of sulfate-deprived rice plants (Figure 2A, Appendix A). The number of DEGs was much more in roots than in the shoots under S deficiency conditions. In the roots, there were 6749 DEGs in total, including 2719 up-regulated and 4030 down-regulated genes (Figure 2A, Appendix A). The DEGs in the shoots and roots accounted for 4.25% (1589/37345) and 18.07% (6749/37345) of the total transcripts detected, respectively.

To understand the functions of these DEGs, we performed GO (Gene Ontology) and KEGG function analysis. GO function analysis was based on the GO database to classify the target genes into three major categories, which were biological process (BP), cellular component (MF), and molecular function (CC). In the shoots, the DEGs were enriched in biological processes and molecular function (Figure 2F); however, the DEGs in the roots were enriched in cellular components (Figure 2G). Among the top 20 significant GO terms (the 20 terms with the lowest *p* values) in shoots, 7 and 11 terms were grouped into the biological processes and molecular function, respectively (Figure 2F). In contrast, 60% of the GO terms were involved in cellular components in the roots (Figure 2G). These results suggested that plants responded differentially to S deficiency in shoots and roots. 

To further dissect the functions of DEGs in response to S deficiency, KEGG analysis was used to detect the pathways in which DEGs were involved. The up-regulated and down-regulated DEGs in shoots or roots were mapped to the public KEGG pathway database, respectively. In the shoots, both up-regulated and down-regulated DEGs were mainly involved in the biosynthesis of secondary metabolites (Figure 2B,C). The biosynthesis of diterpenoid and phenylpropanoid were also enriched in the up-regulated and down-regulated DEGs, respectively (Figure 2B,C). The up-regulated DEGs in roots were related to the biosynthesis of aminoacyl-tRNA and ribosome, while the down-regulated DEGs were enriched in the metabolic pathways (Figure 2D,E). 

In order to investigate the conserved response to S deficiency between shoots and roots, we developed Venn diagrams to identify common DEGs between tissues. Among the 967 up-regulated DEGs in shoots and 2719 up-regulated DEGs in roots, 392 DEGs existed in both shoots and roots, accounting for 40.5% and 14.4% of the up-regulated DEGs in the shoots and roots, respectively. These tissue-common DEGs were mainly enriched in alpha-linolenic acid metabolism and betalain biosynthesis, as revealed by KEGG analysis (Figure 2I). For the down-regulated DEGs, 250 of them were found in both roots and shoots, accounting for 40.2% of up-regulated DEGs in shoots but only 6.2% in roots (Figure 2I). The common down-regulated DEGs were mainly involved in DNA replication (Figure 2I).

### 2.3. Transcriptomic Response of S-deficient Plants to Sulfate Resupply

The expressions of numerous genes were altered in both the shoots and roots of the plants grown under limited sulfate conditions (Figure 2A). In order to explore the transcriptomic response of S-deficient plants to sulfate resupply, we performed a gene expression trend analysis on the expression of DEGs in S-deficient conditions after sulfate resupply for 12 h. These genes can be divided into two groups. One was up- or down-regulated in S deficiency but recovered after sulfate resupply, and the other was not recovered. These two groups of genes were identified by comparing up-regulated DEGs of the shoots or roots under the S-deficient treatment (S0_Sh/CK_Sh up or S0_R/CK_R up) to the genes significantly down-regulated after sulfate resupply (SR_Sh/S0_Sh down or SR_R/S0_R down), or comparison of down-regulated DEGs in the S-deficient treatment (S0_Sh/CK_Sh down or S0_R/CK_R down) with genes significantly up-regulated after sulfate resupply in the shoots or roots (SR_Sh/S0_Sh up or SR_R/S0_R up), respectively. 

Among the 967 up-regulated DEGs and 622 down-regulated DEGs in shoots of sulfate-deprived plants, 219 and 211 DEGs were significantly down-regulated or up-regulated after sulfate resupply, respectively (Figure 3A,B; Appendix A). The number of these expression-recovered DEGs accounts for 22.6% and 33.9% of the total up-regulated or down-regulated DEGs by S deficiency, respectively (Figure 3A,B). The percentage of expression recovered DEGs in the roots was smaller than that in the shoots. Only 15.2% (413/2719) of the up-regulated DEGs and 28.9% (1164/4030) of the down-regulated DEGs in the roots were down-regulated or up-regulated after sulfate resupply, respectively (Figure 3C,D; Appendix A). These results suggested that the expression of a considerable proportion of S-deficiency responsive genes did not recover after sulfate resupply for 12 h (Appendix A).

KEGG network analysis was performed to determine the functions of DEGs with expression recovered or unrecovered by sulfate resupply. In the shoots, the up-regulated DEGs with their expression recovered by sulfate resupply were enriched in the metabolic pathways and biosynthesis of secondary metabolites, including the biosynthesis of phenylpropanoid, flavonoid, and benzoxazinold (Figure 3A). Similarly, the shoots down-regulated DEGs with a recovered expression were also enriched in the metabolic pathways and biosynthesis of secondary metabolites (Figure 3B). The unrecovered up-regulated DEGs after sulfate resupply mainly participated in diterpenoid biosynthesis (Figure 3A). The expression of several genes involved in sulfate uptake, assimilation, and metabolism were recovered after sulfate resupply, including the high-affinity sulfate transporter gene *OsSULTR1;1* (Os03g0195800), putative cysteine synthase gene (Os02g0222100), and cystathionine beta-synthase genes (Os04g0136700, Os10g0499400) (Appendix A). Furthermore, *OsSDI1* (Os03g0165900) and *OsSDI2* (Os05g0506000), the homologs of Arabidopsis S-deficiency marker gene *sulfur deficiency induced 1* (*SDI1*) [28], were strongly induced by S deficiency but their expressions were also suppressed by sulfate resupply (Appendix A). These results suggested that the resupply of sulfate to S-deficient plants for 12 h was able to recover the expression of some S-deficiency responsive genes.

### 2.4. Local and Systemic Response to Sulfate Resupply after S Deficiency in Roots

We took advantage of the split-root system to identify genes that locally or systemically respond to sulfate resupply after S deficiency. Rice plants under 15 d S starvation were equally separated into two halves, and one root half was resupplied with sulfate (SPSR) for 12 h while the other root half remained under sulfate-deprived conditions (SPS0) (Figure 1A). Sulfate was also resupplied to both sides of the roots of the sulfate-deprived plants in the split-root system (SR) as control (Figure 1A). The local and systemic response genes were identified from those S-deficiency responsive genes whose expressions were recovered after sulfate resupply. The local response genes to sulfate resupply were defined as the expression recovered in the split-root half resupplied with sulfate (SPSR) but not in the other root half, which remained under S deficiency. In contrast, the systemic response genes were those genes whose expressions remained unchanged in the sulfate-resupplied root half but were recovered in the S-deficient split-root half. 

To identify the local response genes, we first identified the recovered genes that were induced by S deficiency but downregulated by sulfate resupply or the genes that were downregulated under S deficiency but up-regulated after sulfate resupply. To do this, we overlapped genes in the S0_R/CK_R-up group and the SR_R/S0_R-down group (S0_R/CK_R up ∩ SR_R/S0_R down) (Figure 4A), or genes in the S0_R/CK_R-down group and the SR_R/S0_R-up group (S0_R/CK_R down ∩ SR_R/S0_R up) (Appendix A). The expression-recovered genes were further filtered to include only those genes recovered in the sulfate resupply split-root half but not in the S-deficient split-root half [(S0_R/CK_R up ∩ SR_R/S0_R down ∩ SPSR_R/S0_R down)-SPS0_R/S0_R down; or (S0_R/CK_R down ∩ SR_R/S0_R up ∩ SPSR_R/S0_R up)-SPS0_R/S0_R up]. In total, 128 S deficiency-induced genes were identified as local response genes (Gene sets A in Figure 4A; Appendix A), and 28 local response genes that were suppressed by S deficiency (Gene sets E in Appendix A), accounting for 30.99% (128/413) and 2.41% (28/1164) of DEGs in response to S deficiency but recovered after sulfate resupply, respectively (Appendix A). GO annotation revealed that the local response genes (induced by S deficiency; Gene set A) were mainly related to a cellular amino acid metabolic process and a response to acid chemicals (Figure 4B). As shown in Figure 2D, sulfate deficiency had a significant effect on the metabolism process of various amino acids in the roots. These results implied that the utilization of sulfate in the roots is closely related to the metabolism of amino acids. The S-deficiency-suppressed local response genes (Gene set E) were generally involved in root growth and development, plant organ development and ubiquitin activities (Appendix A).

We used similar strategies to identify the systemic response genes to sulfate resupply after the S deficiency in the roots. The expression-recovered genes were filtered to only include those whose expressions remained unchanged in the sulfate resupply split-root half but recovered in the S-deficient split-root half. To this end, a total of 15 genes were identified as systemic response genes to sulfate resupply, including eight genes that were induced by S deficiency (Gene sets B in Figure 4A; Appendix A) and seven genes that were downregulated under S deficiency (Gene sets F in Appendix A; Appendix A). According to the functional annotations of these systemic response genes, four of them encoded calcium transport and signaling proteins, and three encoded enzymes, including L-lactate dehydrogenase, triacylglycerol lipase, and GDSL-like lipase facylhydrolase (Appendix A).

We further identified the genes that both locally and systemically responded to sulfate resupply after S starvation. The simultaneous local and systemic response genes were isolated from the genes whose expressions were recovered in both the sulfate resupplied split-root half and the S-deficient split-root half. There were 190 simultaneous local and systemic response genes, including 105 S-deficiency-induced genes (Gene sets C in Figure 4A; Appendix A) and 85 genes that were downregulated under S deficiency (Gene sets G in Appendix A; Appendix A). According to the GO annotation, the simultaneous local and systemic response genes in Gene sets C were significantly enriched in the regulation of transcription, while those in Gene sets G were mainly enriched in response to membrane activities, such as an anchored component of the membrane, an intrinsic component of the plasma membrane, and a plasma membrane part (Figure 4B and Appendix A). Interestingly, the sulfate transporter gene *OsSULTR1;1* and S-deficiency-induced marker gene *OsSDI1* were both in the list of simultaneous local and systemic response genes (Appendix A).

Except for the local, systemic, and simultaneous local and systemic response genes in the roots, there were a large proportion of genes that did not respond to sulfate resupply either in the sulfate resupplied split-root half or in the S-deficient split-root half. For the S-deficiency-induced genes that were recovered after sulfate resupply, 41.65% of them (172/413) did not respond to sulfate resupply in either of the split-root halves (Gene sets D in Figure 4A; Appendix A). Moreover, approximately 90% (1044/1164) of the downregulated genes under S deficiency did not respond to sulfate resupply in either of the split-root halves (Gene sets H in Appendix A; Appendix A). GO annotation revealed that no local or systemic response genes were primarily related to translation factor activity and RNA binding in the Gene D sets (Figure 4D) and were related to cell activities in the Gene H sets, including cell periphery, cell cycle, and cell cycle process and Appendix A). The sulfate transporter gene *OsSULTR1;2* (Os03g0196000) and *APK* (Os03g0202001) were identified as not being the local or systemic response genes (Appendix A). The enrichment of the genes that were not local or systemic response genes involved in cell activities suggested that short-term sulfate resupply for 12 h may not be enough to alleviate S deficiency stress.

### 2.5. Transcriptomic Response to Homogeneous and Heterogeneous Sulfate Resupply in Shoots

There were 967 and 622 DEGs that were up- or down-regulated in the shoots of plants under S deficiency, respectively (Figure 2A). Among these DEGs, 219 and 211 of the DEGs were significantly down-regulated or up-regulated, respectively, after homogeneous sulfate resupply in which both of the split-root halves of the S-starved plants were supplied with 0.45 mM SO_4_^2−^ (Figure 1A, Appendix A). Therefore, the homogeneous sulfate resupply recovered the expression of 27.06% (430/1589) of S-deficiency responsive genes. To identify the DEGs that responded to the heterogeneous sulfate resupply in which only one split-root half was supplied with sulfate, but the other half remained S deficient (Figure 1A), we filtered the homogenous sulfate resupply-responding DEGs to include those DEGs with expressions that also recovered in the split-root half that remained S deficient. For the 219 up-regulated DEGs, there were 129 DEGs whose expressions were recovered after heterogeneous sulfate resupply in roots, accounting for 58.9% (129/219) of the expression-recovered genes (Gene set A in Figure 4C; Appendix A). The up-regulated DEGs responding to heterogeneous sulfate resupply were mainly involved in protein translation and peptide biosynthetic process (Figure 4D). In terms of the down-regulated DEGs, there were 101 DEGs with expressions recovered after heterogeneous sulfate resupply, representing 47.42% (101/213) of the expression recovered genes (Gene set C in Appendix A; Appendix A). In total, 20.76% (330/1589) of the S-deficiency responsive genes were recovered by heterogeneous sulfate resupply. GO annotation suggested that these DEGs were enriched in the oxidation-reduction process and cofactor binding (Figure 4D). Several genes involved in sulfur metabolism in the shoots responded to heterogeneous sulfate resupply in the roots, including *APR* (Os07g0509800), *ATPS* (Os04g0111200), and *OAS-TL* (Os06g0564700) (Appendix A).

Approximately 40% (90/219) of the S-deficiency-induced genes in the shoots were only down-regulated after a homogenous sulfate resupply but reminded unchanged in the heterogeneous sulfate resupply (Gene set B in Figure 4C; Appendix A), which mainly contained transporter genes (Figure 4D). Meanwhile, 52.58% (112/213) of the S-deficiency suppressed genes in the shoots were only recovered after a homogenous sulfate resupply but not in the condition of the heterogeneous sulfate resupply (Gene set D in Appendix A; Appendix A). These results suggested that sulfate resupply in one side of the split-root system for 12 h was not able to recover the expression of gene-responding genes to S deficiency in the shoots. 

### 2.6. Response of Genes Involved in Sulfate Uptake and Assimilation to Sulfate Resupply

In order to achieve a global picture of sulfate-related genes in the local and systemic response to heterogeneous S conditions, we summarized the response of the genes involved in sulfate uptake, assimilation, and metabolism in Figure 5. We first collected the response of sulfate transporter genes in the shoots and roots. The *OsSULTR* gene family had 12 members in rice, which displayed different expression patterns under various S conditions. Due to the extremely low expression level in both the roots and shoots under different S conditions (Appendix A), *OsSULTR3;5* was not included in Figure 5. To detect the responses of rice *OsSULTR* genes to S deficiency and sulfate resupply, we compared the FPKM values of *SULTR* genes in different sulfate treatments. In the shoots, the expression of *OsSULTR1;1* was significantly induced by S deficiency, but its expression level did not fully recover after either homogenous or heterogeneous sulfate resupply for 12 h (Figure 5A). The *OsSULTR2;2* was significantly suppressed by S deficiency in the shoots and was not recovered after homogenous or heterogeneous sulfate resupply. Although *OsSULTR2;1*, *OsSULTR3;4*, *OsSULTR3;6*, and *OsSULTR4;1* also responded to S deficiency, their expression differences did not reach a significant level. The remaining *OsSULTR* genes did not significantly respond to S deficiency (Figure 5A). 

In the roots, *OsSULTR1;1* expression was strongly induced by sulfate starvation but significantly down-regulated in both the split-root half with sulfate resupply or the split-root half that remained under S starvation, which suggests both a local and systemic response to sulfate resupply (Figure 5B). Interestingly, *OsSULTR1;2* presented with an opposite expression pattern to *OsSULTR1;1*. Sulfate starvation down-regulated *OsSULTR1;2* expression, and homogenous sulfate resupply up-regulated its expression. However, the expression of *OsSULTR1;2* remained unchanged in either the split-root halves with or without sulfate resupply. These results suggested that *OsSULTR1;2* did not locally or systemically respond to sulfate resupply in roots (Figure 5B). *OsSULTR3;1* was also induced by S deficiency, while the expressions of other rice *SULTRs* were not significantly altered. 

In order to explore the effects of S deficiency, whole-root sulfate resupply, and split-root sulfate resupply on the sulfate assimilation and metabolism, we collected the expression levels of genes involved in the sulfate assimilation pathway (Figure 5C, Appendix A). Several genes were significantly induced by S deficiency, including the *ATPS* (Os03g0743900), *APR* (Os07g0509800), and *OAS-TL* (Os12g0625000) in the shoots and *SHM* (Os12g0409000), and *SAT* (Os03g0196600) in the roots (Figure 5C). Meanwhile, *APK* (Os03g0202001), another member of *OAS-TL* (Os06g0564700) and *SAT* (Os03g0185000) were down-regulated under S deficiency in shoots, and four *OAS-TL* genes (Os06g0564700, Os06g0149700, Os04g0165700, and Os06g0564500) were suppressed in the roots (Figure 5C). Among the five S-deficiency-responsive genes in the shoots, the expressions of three genes were recovered after homogenous or heterogeneous sulfate resupply, including the *APR* (Os07g0509800), *SAT* (Os03g0185000), and *OAS-TL* (Os06g0564700) (Figure 5C; Appendix A). In the roots, the expressions of three of the eight S-deficiency responsive genes were recovered after full root sulfate resupply, including the *APK* gene (Os03g0202001) and two *OAS-TL* genes (Os06g0564700 and Os06g0149700) (Figure 5C; Appendix A). In the biosynthesis pathway of glutathione and methionine, the expressions of glutathione synthase genes (*GSHA* and *GSHB*) and methionine synthesis-related genes (*CGS*, *CBL*, and *MS*) were all unchanged under various S treatments. 

### 2.7. Validation of RNA-Seq by qRT-PCR

To confirm the transcriptome data from RNA-seq, we randomly selected 12 DEGs with diverse responses to S deficiency and sulfate resupply, including six DEGs in shoots and six DEGs in roots. These genes covered *OsSULTR1;1* and *OsSULTR1;2*, as well as *APR* and *ATPS*, which were involved in sulfate uptake and assimilation. The expression patterns of these DEGs were verified through qRT-PCR. The results demonstrated that the qRT-PCR results were in good agreement with the RNA-seq data, which indicated that the RNA-seq results of this study were reliable (Figure 6A,B). 

The qRT-PCR results also confirmed the homogeneous and heterogeneous response to sulfate resupply in the shoots and the local and systemic response in roots. In the shoots, the sulfate-assimilation-related genes *ATPS* (Os04g0111200), *APR* (Os07g0509800), *Methionine synthase* gene (Os12g0624000), and a gene encoding thaumatin (Os07g0417600) all responded to both homogeneous and heterogeneous sulfate resupply (Figure 6A). However, a receptor-like cytoplasmic kinase gene *OsRLCK253* (Os08g0374701), which has been shown to be strongly induced by dehydration and salt stresses [30], only responded to homogeneous but not heterogeneous sulfate resupply (Figure 6A). Interestingly, *DUF1677* (Os02g0198000), encoding a protein of unknown function domain, showed an opposite expression pattern that only responded to heterogeneous but not homogeneous sulfate resupply (Figure 6A). In the roots, *OsSULTR1;1* (Os03g0195800), *OsSDI1* (Os03g0165900), MDR-like ABC transporter gene *OsABCB5* (Os01g0695800), and a protein kinase gene (Os01g0694000) were confirmed to be a local and systemic response to sulfate resupply (Figure 6B). *OsSULTR1;2* (Os03g0196000), however, showed no clear local or systemic response to sulfate resupply (Figure 6B). Another protein of unknown function domain gene *DUF295* (Os01g0660700) responded locally but not systemically to sulfate resupply (Figure 6B). 

## 3. Discussion

S is considered to be the fourth macronutrient, ranking after nitrogen, phosphorus, and potassium, and plays essential roles in plant growth and development and stress resistance. Long-term S deprivation leads to numinous changes in plant morphology and gene expression, including the activation of S-deficiency-responsive genes to uptake adequate sulfate to sustain plant survival [6,22,23]. Once S deficiency has been eliminated, either by homogeneous or heterogeneous sulfate resupply, plants must sense and respond to S resupply in order to recover to the normal growth stage. Such a recovery involves local and systemic responses to sulfate resupply, which is still poorly understood. In this study, we took advantage of the split-root system to investigate the local and systemic response of S-deficient rice plants to sulfate resupply at the transcriptome-wide level. We found that a 15-day sulfate starvation treatment strongly inhibited plant growth and reduced the total shoot or root S concentrations to half of the control plants, suggesting a severe S deficiency stress (Figure 1A–D). For gene expression, a total of 2719 and 4030 genes in the roots and 967 and 622 genes were up- and down-regulated, respectively, under S deficiency (Figure 2A). The number of DEGs in the roots in response to S deficiency was much higher than that in the shoots, indicating a prevalent stress effect on the roots. In the shoots, the expressions of genes involved in the primary sulfate assimilation pathway were generally up-regulated, including the *ATPS* (Os03g0743900), *APR* (Os07g0509800), and *OAS-TL* (Os12g0625000); however, the *APK* gene (Os03g0202001), which catalyzes the biosynthesis of PAPS in the secondary sulfate assimilation branch, was significantly suppressed (Figure 5C). These results suggested that under S-deficient conditions, plants may enhance the primary sulfate assimilation pathway to preferentially synthesize Cys for critical biological processes but inhibit the secondary sulfate assimilation branch, which provides activated sulfate for sulfation reactions. 

Sulfate resupply either on both sides or a single side of the split-root for 12 h did not significantly increase the total S concentrations in the roots or shoots of rice plants under a 15 d S deficiency stress (Figure 1C,D), suggesting that a long-term S deficiency leads to a serious supply shortage of S, and a 12 h sulfate resupply was not enough to completely alleviate the growth stress caused by S deficiency. Under the treatments with a long-term sulfate deficiency, multiple metabolic pathways were suppressed, including sulfate assimilation and cysteine metabolism, but such suppressions were alleviated to a certain extent after sulfate resupply (Figure 3A–D). These results indicated that the plant might preferentially synthesize various S-containing compounds through the sulfate assimilation pathway to support plant growth. Similarly, S-deficient Arabidopsis plants could rapidly synthesize S-containing amino acids such as Cys and GSH, which exceeded the control level within 3 h of sulfate resupply [26]. 

Although a 12 h sulfate resupply was not able to significantly increase the total S level in the roots or shoots of S-deficient plants (Figure 1C,D), the expressions of a considerable proportion of S-deficiency-responsive genes were recovered after sulfate resupply (Figure 3A–D). Previous studies in Arabidopsis have identified many genes involved in the local and systemic regulation of S homeostasis by transcriptomic analysis [22,31,32]. By determining the recovery of S-deficiency-responsive genes in the split-roots, we identified four types of gene sets in response to sulfate resupply, including local response, systemic response, simultaneous local and systemic response, and no local or systemic response. Although the non-local or non-systemic response genes accounted for a large proportion of expression-recovered genes, our results clearly showed that both local and systemic responses existed in S-deficient rice plants in response to sulfate resupply. Regarding the local response genes, which only responded in the split-root half with sulfate resupply but not in the other split-root half that remained deficient in S, most of them were involved in cellular amino acid metabolic processes and root growth and development (Figure 4B and Appendix A), suggesting a 12 h of sulfate resupply in one half of the root was not able to trigger the reprogramming of root growth in the other half without sulfate resupply.

Membrane proteins that fulfill the dual function of nutrient transport and nutrient-sensing have been termed ‘transceptors’ [33,34,35,36,37]. In yeast, sulfate transporters SUL1 and SUL2 have been shown to function as sulfate transceptors with both sulfate transport activities and are able to transduce the S signal to activate the PKA signaling pathway and trigger downstream biological processes [32]. Arabidopsis AtSULTR1;2 may also function as a sulfate transceptor, as the defect of the S signaling in the *atsultr1;2* mutants is independent of sulfate transport and accumulation [31,38]. The expressions of *AtSULTR1;2* and *AtSULTR1;1* were strongly induced by S deficiency, and such an induction apparently only occurred in the S-depleted split-root half but not in the other root half supplied with S, suggesting a local but not systemic response to heterozygous S deficiency [19]. In this study, we found that the responses of rice *OsSULTR1;1* and *OsSULTR1;2* to sulfate resupply after S deficiency were distinct from the pattern of *AtSULTR1;2* and *AtSULTR1;1* in response to heterozygous S deficiency. *OsSULTR1;1* was strongly induced by sulfate starvation in the roots and significantly down-regulated in both the split-root half with sulfate resupply and the split-root half that remained under S starvation, suggesting both a local and systemic response of *OsSULTR1;1* to sulfate resupply (Figure 5B). However, in contrast to the strong induction of *AtSULTR1;2* by S deficiency [1], the expression of *OsSULTR1;2* was suppressed under sulfate starvation conditions (Figure 5B). Furthermore, the expression of *OsSULTR1;2* remained unchanged in both the split-root halves with or without sulfate resupply, suggesting no local or systemic response to sulfate resupply in roots (Figure 5B). The different responses to variable S status of *SULTR1;1* and *SULTR1;2* in rice and Arabidopsis suggested their distinct functions in controlling S homeostasis in plants, which requires further studies.

In addition to *OsSULTR1;1*, the S-deficiency-induced marker gene *OsSDI1*, which encoded a tetratricopeptide-like helical domain-containing protein, also showed simultaneous local and systemic response to sulfate resupply (Appendix A). Arabidopsis SDI1, the homolog of OsSDI1, acted as a major repressor in controlling glucosinolates (GSLs) biosynthesis under S-limited conditions [39]. SDI1 formed a protein complex with MYB28 transcriptional factor and inhibited the transcription of genes involved in aliphatic GSL biosynthesis, and prioritized the sulfate usage for primary metabolites under sulfur-deprived conditions [39]. The biosynthesis of GSLs did not exist in rice, so the function of OsSDI1 was not clear. Given that *OsSDI1* was strongly induced by S deficiency and behaved with a local and systemic response to sulfate resupply, it is similar to that of OsSDI1 interacting with MYB transcriptional factor(s) and plays a key role in the prioritization of sulfate usage under S-limited conditions. Consistent with this speculation, two MYB transcription factor genes, *OsMYB71* (Os09g0431300) and *OsMYB4* (Os01g0695900), were found to respond locally and systemically to sulfate resupply in the roots (Appendix A). The beta-glucosidase gene *BGLU28* is one of the most strongly induced genes by S deficiency in Arabidopsis and was thought to function in the breakdown of GSLs to release S for plant growth under S-limited conditions [9,25,40,41]. Interestingly, the rice *beta-glucosidase 21* gene (*OsBGLU21*; Os05g0366000) was suppressed by S deficiency and recovered in both split-root halves after sulfate resupply (Appendix A), suggesting its distinct roles from *BGLU28* in response to S deficiency and sulfate resupply.

Compared to a large number of local response genes, only 15 genes were identified as a systemic response to sulfate resupply (Appendix A). Among these systemic response genes, three of them encode calmodulin-binding proteins (Appendix A, Gene set B), and one encodes a calcium-transporting ATPase (Appendix A, Gene set F). Calcium ions are the most prominent second messengers that integrate extracellular signals with specific intracellular responses in plants [42,43]. The calcium signal has been shown to play an important signaling role in systemic plant defense and wound signaling [44,45]. Therefore, it is likely that calcium signaling is involved in the systemic response to sulfate resupply after S deficiency. Two of these three calmodulin-binding proteins (Os02g0305950 and Os08g0534950) were also annotated as auxin-responsive proteins. Os02g0305950 (*OsSAUR7*) belongs to the *Small Auxin-Up RNA* gene family, which is induced by exogenous auxin within minutes and has been shown to play important roles in diverse processes of plant development and stress responses [46,47]. The homeobox-leucine zipper transcriptional factor gene *OsHOX22* (Os04g0541700) responded to sulfate resupply systemically in the split-root without sulfate resupply but not in the local split-root with sulfate resupply (Appendix A, Gene set B). *OsHOX22* is involved in the abscisic acid (ABA)-mediated drought and salt tolerances in rice [48]. Os02g0668500, which encodes a Rho GTPase-activating protein, also systemically responded to sulfate resupply (Appendix A, Gene set B). The Rho of plants (ROP) GTPase signaling network converges on a wide range of upstream signals and elicits downstream signaling cascades to modulate developmental processes, including the auxin and ABA signaling [49,50,51]. The enrichment of systemic response genes in auxin and ABA signaling suggested that plant hormones auxin and ABA may participate in the systemic response of sulfate resupply. This is supported by previous studies that auxin plays a negative role in the regulation of S deficiency response in Arabidopsis [40]. Further studies are required to investigate the roles of auxin and ABA in the regulation of systemic responses to S status.

## 4. Materials and Methods

### 4.1. Plant Materials and Growth Conditions

The rice (*Oryza sativa* L.) cultivar Zhonghua 11 was used in this experiment. The seeds were germinated in an incubator at 37 °C for 3 d, then sowed on a plastic net floating on ultra-pure deionized water. After 7 d, the seedlings were transplanted into plastic boxes containing 1/2 Kimura nutrient solution (0.27 mM of MgSO_4_, 0.18 mM of (NH_4_)_2_SO_4_, 0.18 mM of Ca(NO_3_)_2_, 0.09 mM of KNO_3_, 90 mM of KH_2_PO_4_, 20 μM of NaEDTAFe, 3 μM of H_3_BO_3_, 0.5 μM of MnCl_2_, 0.2 μM of CuSO_4_, 0.4 μM of ZnSO_4_, and 0.01 μM of (NH4)_6_Mo_7_O2_4_; pH, 5.6). The plants were grown in a glasshouse supplemented with 250 μmol m^−2^ s^−1^ light in a 12-h-light (30 °C)/12-h-dark (24 °C) photoperiod and approximately 60% relative humidity. The nutrient solution was renewed every 3 d. When the fifth leaf fully emerged, the nutrient solution was replaced with a modified 1/2 Kimura nutrient solution in which the 0.27 mM of MgSO_4_, 0.18 mM of (NH_4_)_2_SO_4_, 0.2 μM of CuSO_4_, and 0.4 μM of ZnSO_4_ were replaced by 0.27 mM of MgCl_2_, 0.36 mM of NH_4_Cl, 0.2 μM of CuCl_2_ and 0.4 μM of ZnCl_2_, respectively. The S deficiency treatment was performed for 15 d. A set of plants was kept growing in standard 1/2 Kimura nutrient solution with 0.45 mM of SO_4_^2−^ sulfate as the control. 

To split the roots, the roots of the sulfate-deficiency treated plants were gently separated into approximately equal parts on the 13th day after the sulfate deficiency treatment. The split-root plants were then transferred to a split root system with two separate compartments and kept growing in sulfate deficiency conditions for another 2 d to reduce the effect of root splitting on plant growth (Figure 1A). The split root system was generated by separating a 10-L culture box with a sealing clapboard in the middle to prevent the diffusion of nutrients between the left and right sides. After 15 d of treatment, the split-root plants were recovered with standard 1/2 Kimura nutrient solution containing 0.45 mM SO_4_^2−^ either on both sides or on one side only. The sulfate resupply treatment was performed for 12 h before harvesting samples for RNA-seq or determining total S concentrations.

### 4.2. Sulfur Concentration Determination

After various sulfate treatments, the plants were washed three times with ultrapure water, and the shoots and roots were harvested separately and briefly dried with tissue paper. The plant samples were dried at 65 °C for 3 d and then digested with 2 ml of HNO_3_ at 120 °C for 4 h. The blank tubes without samples and the control tubes with standard samples were digested under the same conditions. Spinach leaves were used as standard samples for the plant samples (GBW10015, Institute of Geophysics and Geochemical Exploration, Langfang, China). The S concentration was determined using an inductively coupled plasma mass spectrometry (ICP-MS, NexION 300X, PerkinElmer, Waltham, MA, USA).

### 4.3. RNA Extraction, cDNA Library Construction and Transcriptome Sequencing

The total RNAs were extracted using a BioTeke Plant total RNA Extraction Kit (Bioteke, Beijing, China). The RNA quality was assessed on an Agilent 2100 Bioanalyzer (Agilent Technologies, Palo Alto, CA, USA) and checked using RNase-free agarose gel electrophoresis. After the total RNA was extracted, eukaryotic mRNA was enriched by Oligo(dT) beads, while prokaryotic mRNA was enriched by removing the rRNA with a Ribo-ZeroTM Magnetic Kit (Epicentre, Madison, WI, USA). Then the enriched mRNA was fragmented into short fragments using a fragmentation buffer and reverse transcript into cDNA with random primers. The second-strand cDNA was synthesized by DNA polymerase I, RNase H, dNTP, and buffer. Then the cDNA fragments were purified with a QiaQuick PCR extraction kit (Qiagen, Venlo, The Netherlands), end-repaired, poly(A) added, and ligated to Illumina sequencing adapters. The ligation products were size selected by agarose gel electrophoresis, PCR amplified, and sequenced using Illumina HiSeq2500.

### 4.4. RNA-Seq Data Analysis

To achieve high-quality clean reads, the reads were further filtered by the fastp tool (version 0.18.0; https://github.com/OpenGene/fastp; accessed date: 23 April 2020) with the following parameters: (1) removing reads containing adapters; (2) removing reads containing more than 10% of unknown nucleotides (N); (3) removing low quality reads containing more than 50% of low quality (Q-value ≤ 20) bases. Then paired-end clean reads were mapped to the rice reference genome (version IRGSP-1.0; https://rapdb.dna.affrc.go.jp/; accessed date: 23 April 2020) using HISAT2. 2.4 with “-rna-strandness RF” and other parameters set as a default. The mapped reads of each sample were assembled by using StringTie v1.3.1 in a reference-based approach. For each transcription region, an FPKM (fragment per kilobase of transcript per million mapped reads) value was calculated to quantify its expression abundance and variations using StringTie software [52].

### 4.5. Principal Component Analysis (PCA) 

PCA was performed using R package gmodels (R3.6.1; http://www.rproject.org/; accessed date: 30 April 2020). Principal component analysis (PCA) is a statistical procedure that converts hundreds of thousands of correlated variables (gene expression) into a set of values of linearly uncorrelated variables called principal components. PCA is largely used to reveal the relationship of the samples. 

### 4.6. Identification of DEGs

RNAs differential expression analysis was performed by DESeq2 software between two different groups or by edgeR between two samples. The genes/transcripts with the parameters of false discovery rate (FDR) < 0.01, *p* value < 0.05 and absolute fold change ≥ 2 were considered differentially expressed genes/transcripts.

### 4.7. GO and KEGG Analysis

The assembled genes and novel transcripts were annotated according to public databases: Gene Ontology (GO) and Kyoto Encyclopedia of Genes and Genomes (KEGG). GO enrichment analysis provides all of the GO terms that were significantly enriched in DEGs compared to the genome background and filters the DEGs that correspond to biological functions. The GO enrichment analysis was performed using the OmicShare tools, a free online platform for data analysis (www.omicshare.com/tools; accessed date: 3 May 2020). Firstly, all of the DEGs were mapped to GO terms in the Gene Ontology database (http://www.geneontology.org/; accessed date: 3 May 2020). The gene numbers were calculated for every term, and significantly enriched GO terms in DEGs compared to the genome background were defined by a hypergeometric test. The calculated *p* value underwent FDR correction, taking FDR ≤ 0.05 as a threshold. The GO terms meeting these criteria were defined as significantly enriched GO terms in DEGs. This analysis was able to reveal the main biological functions of DEGs. 

KEGG is a major public pathway-related database. Pathway enrichment analysis identified significantly enriched metabolic pathways or signal transduction pathways in DEGs compared with the whole genome background. Pathway enrichment analysis was performed using the OmicShare tools (www.omicshare.com/tools; accessed date: 4 May 2020). Significantly enriched pathways in DEGs compared to the genome background were defined by a hypergeometric test. The calculated *p* value was gone through FDR correction, taking FDR ≤ 0.05 as a threshold. Pathways meeting these criteria were defined as significantly enriched pathways in DEGs. Only the top 10 significant pathways of the KEGG pathway and the KEGG network were shown.

### 4.8. Trend Analysis and Venn Diagram Analysis

Gene expression trend analysis was performed by Short Time-series Expression Miner software (STEM). The parameters were set up as follows: (1) Maximum Unit Change in model profiles between time points is 1; (2) Minimum ratio of the fold change of DEGs was no less than 2.0. Venn diagrams were drawn by Biovenn (http://www.biovenn.nl/index.php; accessed date: 10 May 2020), and the heat maps were drawn using R3.6.1 based on FPKM values, and Log2/Log10(FPKM) homogenization was performed for data.

### 4.9. Validation of DEGs by qRT-PCR 

To verify the RNA-seq data, qRT-PCR experiments of 12 randomly selected DEGs were performed. The total RNA concentrations were measured using NanoDrop 2000 (ThermoFisher, Waltham, MA, USA). One microgram of the total RNA was used to synthesize cDNA using a HiScript III RT SuperMix kit for qPCR (+gDNA wiper) (Vazyme, Nanjing, China). The qPCR was conducted on the CFX Manager thermal cycler (Bio-Rad, CA, USA) using SYBR Green Master Mix (Vazyme, Nanjing, China). The program was 95 °C for 30 s, followed by 39 cycles of 95 °C for 10 s and 60 °C for 30 s. The data were presented as 2^−∆Ct^ with rice *Actin* as an internal reference gene. All of the samples were analyzed in triplicate. Primers used for qRT-PCR were listed in Appendix A.

### 4.10. Statistical Analysis

The software SPSS Statistics 25 was used for statistical data analysis, using analysis of variance followed by comparisons of means using Tukey–Kramer test. Bar diagrams and scatter plots with bar diagrams were drawn with GraphPad Prism 8.0.1.

## 5. Conclusions

In this study, by taking advantage of a split-root system and transcriptome-wide gene expression analysis, we showed the existence of local and systemic response to sulfate resupply after S deficiency in rice. At the transcriptome-wide level, a relatively long term S deficiency treatment (15 d) altered the expressions of 18.07% and 4.28% of the total transcripts detected in the roots and shoots, respectively. Homogeneous sulfate resupply in both the split-root halves and heterogeneous sulfate resupply in only one split-root half recovered the expression of 27.06% and 20.76% of S-deficiency-responsive genes in the shoots, respectively. In the roots, 58.35% of S-deficiency-induced genes and approximately 10% of downregulated genes responded to sulfate resupply. Among these sulfate resupply responsive genes, we identified 156 locally responsive genes (128 S deficiency-induced genes and 28 S deficiency downregulated genes) whose expressions were only recovered in the split-root half resupplied with sulfate but not in the other root half, which remained under S deficiency. The local response genes were mainly enriched in the amino acid metabolic process and root growth and development. We also identified 15 systemic response genes whose expressions were recovered in the S-deficient split-root half but not in the sulfate-resupplied root half. The systemic responsive genes were mainly involved in calcium transport and signaling processes, such as the calmodulin-binding proteins and a calcium-transporting ATPase, highlighting the important role of calcium signaling in mediating systemic responses to sulfate resupply. A large number of genes that displayed simultaneous local and systemic responses to sulfate resupply were also identified, including *OsSULTR1;1* and *OsSDI1*. By summarizing the response of genes involved in sulfate uptake, assimilation, and metabolism pathways, we demonstrated that a finely-tuned regulation response to sulfate resupply at the local and systemic levels is essential for plants to recover to normal growth level after S deficiency. Therefore, our studies provide a transcriptome-wide picture of the local and systemic responses of S-deficient rice plants to sulfate resupply, which will deepen the understanding of systemic regulation of S homeostasis in rice.

## Figures and Tables

**Figure 1 ijms-23-06203-f001:**
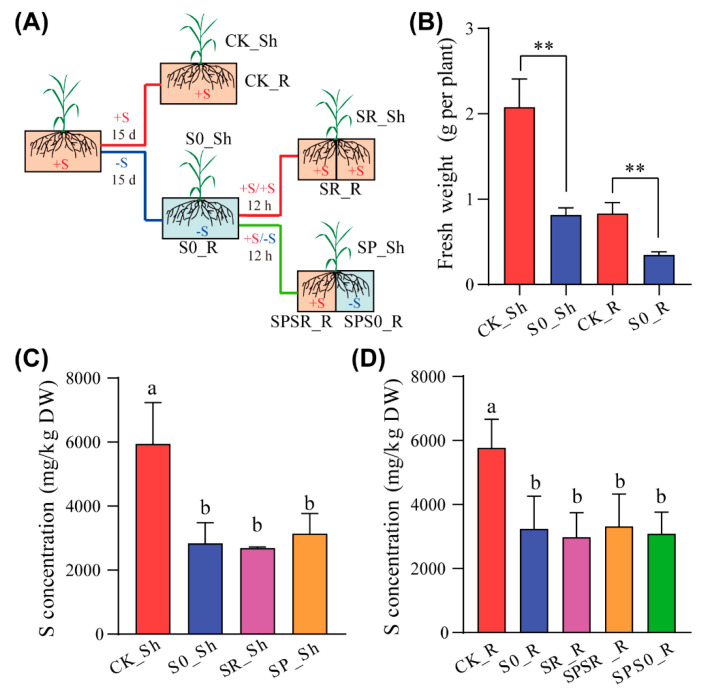
Experimental design and total S concentrations in plants. (**A**) Schematic diagram of experimental design. Plants grown with sufficient sulfate (0.45 mM SO_4_^2−^) to five-leaf stage were subjected to sulfate deficient treatment (0 mM SO_4_^2−^; S0) for 15 d or continued growing with sufficient sulfate as control (CK). The roots of sulfate deficient treated plants were equally split into two halves and then resupplied with 0.45 mM SO_4_^2−^ for 12 h in both split-root halves (SR), or only one split-root halve (SPSR) with the other root half remained in sulfate deficiency (SPS0). (**B**) The fresh weight of roots and shoots of seedling subjected to sulfate deficient treatment for 15 d. (**C**,**D**) Total S concentrations in shoots (**C**) and roots (**D**) of plants resupplied with 0.45 mM SO_4_^2−^ for 12 h in both split-root halves or one split-root halve only. Data in (**B**–**D**) are presented as means ± SD with three biological replicates. ** in (**B**) represents significant differences at *p* < 0.01 (Students’ *t*-test). Columns with different letters in (**C**,**D**) indicate significant difference at *p* < 0.05 (Tukey-Kramer Test). DW, dry weight.

**Figure 2 ijms-23-06203-f002:**
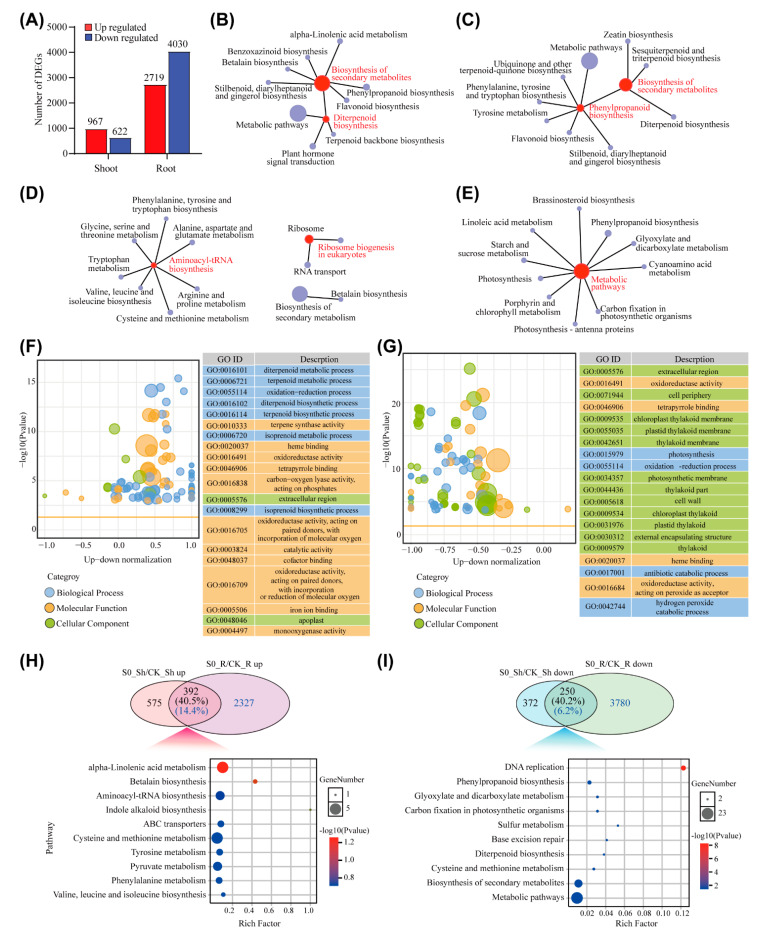
Transcriptomic response to S deficiency in rice. (**A**) Total numbers of DEGs in shoots and roots of plants grown under S-deficient conditions for 15 d. DEGs were identified as absolute fold change ≥ 2 with FDR < 0.05 and *p* < 0.05. (**B**,**C**) KEGG networks of up-regulated DEGs (**B**) and down-regulated DEGs (**C**) in shoots. (**D**,**E**) KEGG networks of up-regulated DEGs (**D**) and down-regulated DEGs (**E**) in roots. (**F**,**G**) Z-score bubble plot of GO enrichment analysis of DEGs in shoots (**F**) and roots (**G**). The vertical axis is −log10(*p* value), and the horizontal axis is the proportion of the difference between the up-regulated DEGs number and down-regulated DEGs number in the total DEGs. The bubble size represents the number of DEGs enriched in each GO term. The orange line represents the threshold of *p* value = 0.05. The top 20 terms/pathways with lowest *p* value were listed in the table on right. Different colors of bubbles represent different categories. (**H**,**I**) The number and GO analysis of up-regulated DEGs (**H**) or down-regulated DEGs (**I**) in both shoots and roots.

**Figure 3 ijms-23-06203-f003:**
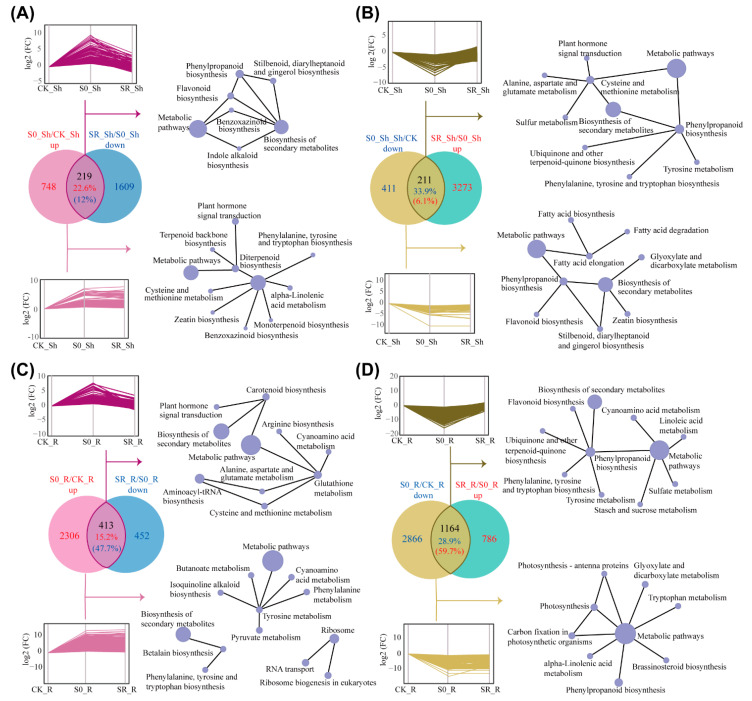
Transcriptomic response of S-deficient plants to sulfate resupply. (**A**,**B**) KEGG network analysis of up-regulated (**A**) or down-regulated DEGs (**B**) in response to sulfate resupply in shoots. (**C**,**D**) KEGG network analysis of up-regulated (**C**) or down-regulated DEGs (**D**) in response to sulfate resupply in roots. The number of DEGs with expression recovered after sulfate resupply were shown in Venn diagrams. The gene expression trend of DEGs with expressions recovered or unchanged in response to sulfate resupply was shown in upper and lower panels, respectively. The size of nodes in KEGG network represents the number of genes. CK, control; S0, S deficiency; SR, sulfate resupply; Sh, shoot; R, root; FC, fold change.

**Figure 4 ijms-23-06203-f004:**
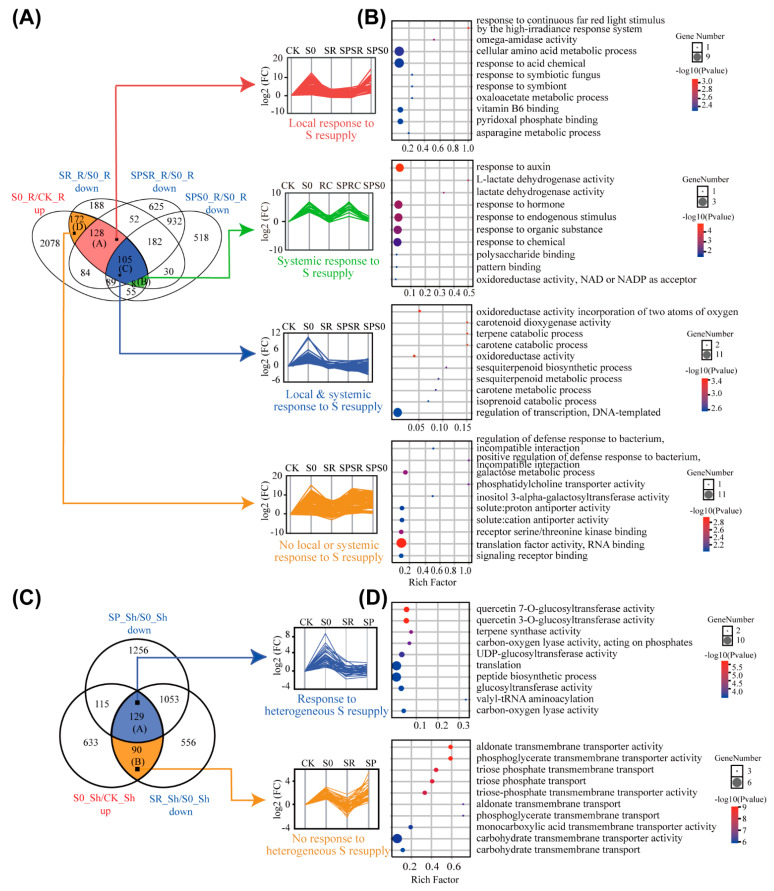
Root and shoot up-regulated DEGs in response to homogeneous and heterogeneous sulfate resupply. (**A**) Identification of root up-regulated DEGs with local response, systemic response, simultaneous local and systemic response, and no response to sulfate resupply by gene expression trend analysis and Venn diagramming. Different types of response genes were classified into Gene set A, B, C, and D, respectively, as shown in Appendix A. (**B**) GO enrichment analysis of different types of response genes in (**A**). The top 10 GO terms with lowest *p* values were shown. (**C**) Identification of shoot up-regulated DEGs responding to homogeneous and heterogeneous sulfate resupply, which were classified into Gene set A and B as shown in Appendix A. (**D**) GO enrichment analysis of different types of response genes in (**C**). The top 10 GO terms with lowest *p* values were shown. CK, control; S0, S deficiency; SR, sulfate resupply in both sides of split-roots; SPSR, the split-root halve with sulfate resupply; SPS0, the split-root halve remained in S deficiency. Sh, shoot; R, root.

**Figure 5 ijms-23-06203-f005:**
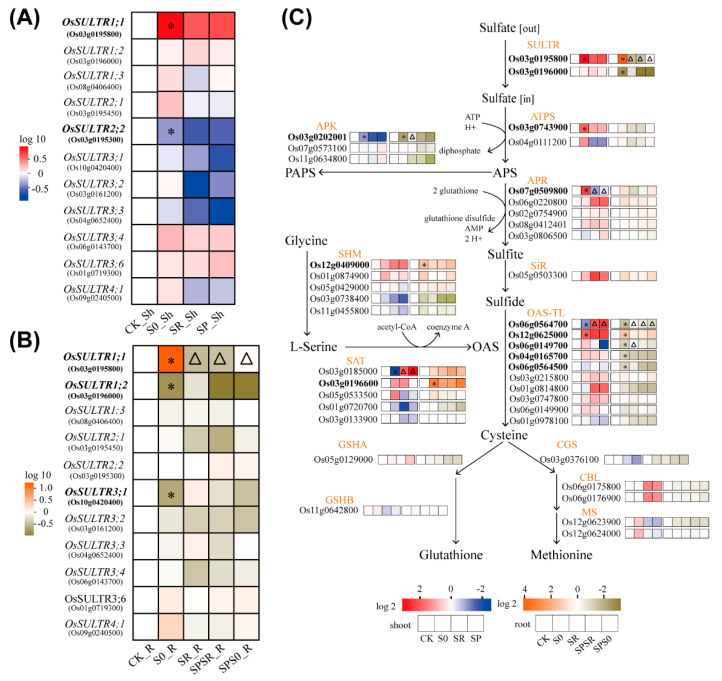
Summary of expression pattern of genes involved in sulfate uptake, assimilation, and metabolism in roots and shoots of plants under various S conditions. (**A**) Relative expression level of sulfate transporter genes in shoots of rice plants under S deficiency or resupplied with sulfate homogeneously or heterogeneously in roots. (**B**) Relative expression level of sulfate transporter genes in roots of rice plants under different S conditions. (**C**) Relative expression level of genes involved in sulfate assimilation and metabolism in roots (right panel) and shoots (left panel) of plants under different S conditions. Relative expression levels in (**A**–**C**) were normalized to the control (CK). Asterisks (*) represented genes that were significantly differentially expressed under S deficiency. Triangles (∆) represented genes showing local, systemic, or both response to sulfate resupply. SULTR: sulfate transporter; ATPS: ATP sulfurylase; APR: adenosine-5′-phosphosulfate reductase; APK: adenosine-5′-phosphosulfate (APS) kinase; SIR: sulfite reductase; OAS-TL: O-acetylserine (thiol) lyase; SAT: serine acetyltransferase; SHM: serine hydroxymethyltransferase; GSHA: gamma-glutamylcysteine synthetase; GSHB: glutathione synthetase B; CGS: cystathionine gamma-synthase; CBL: cystathionine beta-lyase; MS: methionine synthase. CK, control; S0, S deficiency; SR, sulfate resupply in both sides of split-roots; SPSR, the split-root halve with sulfate resupply; SPS0, the split-root halve remained in S deficiency. Sh, shoot; R, root.

**Figure 6 ijms-23-06203-f006:**
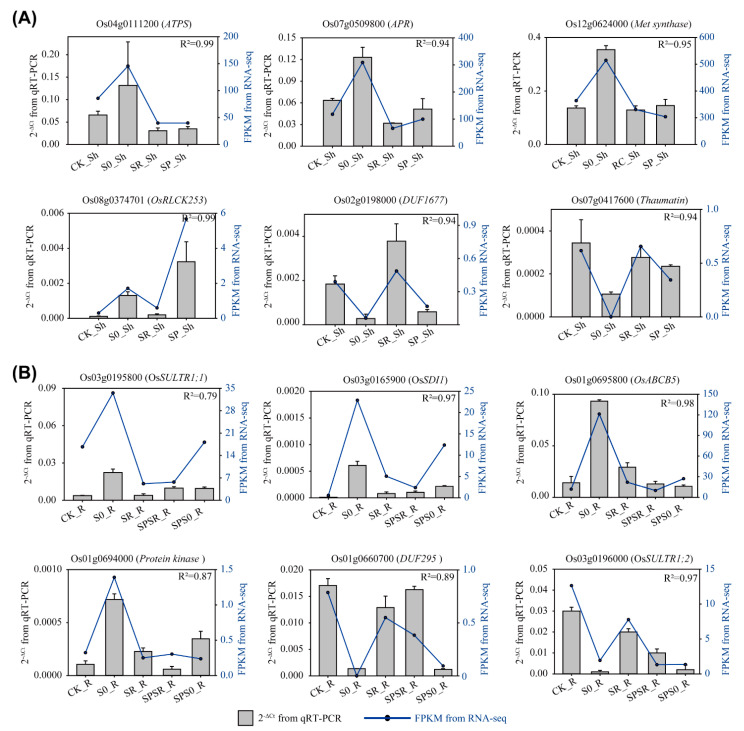
qRT-PCR verification of the expression pattern of selected DEGs in response to sulfate resupply. (**A**) The relative expression levels determined by qRT-PCR and the FPKM values from RNA-seq of six genes in shoots. (**B**) The relative expression levels determined by qRT-PCR and the FPKM values from RNA-seq of six genes in roots. The relative expression levels determined by qRT-PCR were presented as 2^−∆Ct^ using the rice *Actin* gene as internal reference gene. The gray columns indicated the 2^−∆Ct^ values from qRT-PCR, and blue lines represented FPKM values from RNA-seq. Pearson correlation coefficient (R^2^) between 2^−∆Ct^ and FPKM value was calculated using SPSS 22.0. CK, control; S0, S deficiency; SR, sulfate resupply in both sides of split-roots; SPSR, the split-root halve with sulfate resupply; SPS0, the split-root halve remained in S deficiency. Sh, shoot; R, root.

## Data Availability

The RNA-seq datasets were deposited in the Genome Sequence Archive in National Genomics Data Center, China National Center for Bioinformation/Beijing Institute of Genomics, Chinese Academy of Sciences (https://ngdc.cncb.ac.cn/gsa/ (accessed on 20 April 2022)) with accession number provided upon request.

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
