# Peer review of "Local and Systemic Response to Heterogeneous Sulfate Resupply after Sulfur Deficiency in Rice"

_ijms, 2022, doi:10.3390/ijms23116203_

Round 1
Reviewer 1 Report
The current manuscript “Local and systemic response to heterogeneous sulfate resupply after sulphur deficiency in rice” sets out to understand the types of genes involvement in the sulphur deficiency/control/resupply conditions, a major macronutrient involved in the plants growth and defence responses. The manuscript describes in great details about various unique /common set of genes involved in the different types of sulphur supply. I have a following comments regarding the manuscript:
- The title says local and systemic response, however based on the results, it does not provide any contrasting difference between local and systemic response of the DEGs involvement, and therefore, I would suggest to change the title.
- Why only one time point of 12 hour sulphur resupply was chosen? Is there any special rationale behind it?
- Why 0.45mM of sulphate ion was chosen, is it a genotype dependent concentration?
- In the results section of Experimental design and RNA-seq data summary, the statement “Sulfate resupply was applied the sulfate deprived” is grammatically incorrect. Please modify it.
- The labels in the figures 2b-e are confusing. Please make it clear, which are the upregulated/downregulated DEGs involved in roots and shoots.
- We do not find much in the discussion or in the results, the basis of explanation that why
- S0_Sh, SR_Sh and SP_Sh (Figure 1C)
- S0_R, SR_R, SPSR_R, SPS0_R (Figure 1D)
did not differ in their Sulphur concentration
- How the split root system was generated? Many of the non-plant physiologist may not understand it, please explain it in detail.
- Authors have emphasized more on the DEG expression in sulphate resupply and non-supply condition using split root/ whole root supply and deficiency of sulphur in the V5 stage of rice. It would be great to see, if authors can describe a proposed mechanism based on the comparative analysis.
- The calcium signalling related genes upregulation were also observed across four different types of transcriptome comparison. This is quite expected, as Calcium and sulphur plays an important role in the abiotic stress signalling. I am wondering, if the calcium signalling related genes were calcium dependent protein kinase or MAPK related genes? It has been observed in legumes, under calcium and sulphur deficiency, the calcium dependent protein kinase gets upregulated, and remains relatively lowly expressed, when calcium is resupplied.
Reviewer 2 Report
Review of the paper entitled “Local and systemic response to heterogeneous sulfate resupply after sulfur deficiency in rice” by Ru-Yuan Wang, Li-Han Liu, Fang-Jie Zhao, and Xin-Yuan Huang
Sulfur is an important component necessary for both plant and animal life. Plants take up this nutrient in the form of the sulfate ion (SO42-) through the roots and then it is distributed via vascular bundles throughout the plant. Cysteine is the end product of the sulfur assimilation and the major sulfur donor in biosynthetic processes of various sulfur-containing small biomolecules, such as glutathione (GSH), thiamine, biotin, coenzyme A (CoA), lipoic acid (LA), iron-sulfur cluster (Fe/S), molybdenum cofactor (Moco) and sulfur-modified tRNA. Lack of sulfur causes inhibition of plant growth due to disruption of protein synthesis and, consequently, loss of quantity and quality of the obtained crop. As the Authors note, plants have developed sophisticated mechanisms to modify gene expression and physiological processes in order to optimize sulfur acquisition and usage.
In the present paper the Authors used a split root system to analyze gene expression throughout the transcriptome on sulfur-deficient rice plants and then supplemented with sulfate. The obtained results indicated that sulfur deficiency altered the expression of 6,749 and 1,589 genes in roots and shoots, accounting for 4.28% and 18.07% of all transcripts detected, respectively. The Authors showed also that the resupply of sulfate to sulfur deficient plants for 12 h was able to recover the expression of some sulfur deficiency responsive gene.
The paper is interesting and well written.
My comment
I would like to ask the Authors to supplement their manuscript with such elements as:
- a scheme showing the sulfur cycle in general.
Water-soluble sulfates are the main form of sulfur found in nature and are the main source of sulfur available to plants (for a review see: Lewandowska M, Sirko A. Recent advances in understanding plant response to sulfur-deficiency stress. Acta Biochim Pol. 2008;55(3):457-71. Epub 2008 Sep 12. PMID: 18787711). In plant cells, sulfates are reduced to thiol-containing compounds (R-SH), which in turn are the main source of sulfur available to animals and humans. Sulfates in plant cells can also be reduced to hydrogen sulfide and sulfane sulfur compounds (for a review see: Nakai Y, Maruyama-Nakashita A. Biosynthesis of Sulfur-Containing Small Biomolecules in Plants. Int J Mol Sci. 2020 May 14;21(10):3470. doi: 10.3390/ijms21103470. PMID: 32423011; PMCID: PMC7278922).
- Brief summary of the obtained results. Conclusion. Highlights.
- Information on visual symptoms of sulfur deficiency in the plants tested. It is known that in case of insufficient sulfur supply in plants, symptoms characteristic of reduced photosynthetic activity of plants appear. The leaves of the sulfur-deficient plants are small, they are shortened and narrowed. In younger leaves chlorophyll decomposes, as a result they turn light green and yellow. Have the Authors observed such changes? I would like the Authors to consider including a photo of a control plant and a sulfur-deficient plant in their manuscript.

Author Response
Please see the attachment。
